# Advances in Conjugated Polymer Lasers

**DOI:** 10.3390/polym11030443

**Published:** 2019-03-07

**Authors:** Hongyan Xia, Chang Hu, Tingkuo Chen, Dan Hu, Muru Zhang, Kang Xie

**Affiliations:** Dongyuan Synergy Innovation Institute for Modern Industries of GDUT, Guangdong University of Technology, Guangzhou 510006, China; hyxia@gdut.edu.cn (H.X.); huchang689566@163.com (C.H.); chentingkuo@163.com (T.C.); 18390609411@163.com (D.H.); 17665450411@163.com (M.Z.)

**Keywords:** conjugated polymer, laser, amplified spontaneous emission, photoluminescence

## Abstract

This paper provides a review of advances in conjugated polymer lasers. High photoluminescence efficiencies and large stimulated emission cross-sections coupled with wavelength tunability and low-cost manufacturing processes make conjugated polymers ideal laser gain materials. In recent years, conjugated polymer lasers have become an attractive research direction in the field of organic lasers and numerous breakthroughs based on conjugated polymer lasers have been made in the last decade. This paper summarizes the recent progress of the subject of laser processes employing conjugated polymers, with a focus on the photoluminescence principle and excitation radiation mechanism of conjugated polymers. Furthermore, the effect of conjugated polymer structures on the laser threshold is discussed. The most common polymer laser materials are also introduced in detail. Apart from photo-pumped conjugated polymer lasers, a direction for the future development of electro-pumped conjugated polymer lasers is proposed.

## 1. Introduction

“Laser” is an abbreviation of light amplification by stimulated radiation; this technology has been used in almost all industries as well as in the military, technology, entertainment, communications, and other fields since the 1960s [1]. The laser system is mainly composed of a laser gain medium (laser material) which is capable of amplifying light, an excitation (pump) system, and an optical resonator. The laser gain medium is the substance used to realize particle number inversion and generate the stimulated emission of light—it provides the basis for the realization of the stimulated emission and is the core part of the laser system [2]. In the development of laser technology, the development of laser gain materials has played a very important role. A variety of different materials have been explored and tested as the laser gain material. Laser working medium materials can be divided into the following categories: solid gain media, gas gain media, liquid gain media, inorganic semiconductors, etc. Semiconductor lasers have attracted much attention because of their small size, fast response, low power consumption, and high efficiency compared with other types of lasers [3]. With the continuous advancement of materials science, people are no longer limited to the research of inorganic semiconductor laser materials, and organic semiconductor lasers are attracting more and more attention.

In the decade after the birth of the world’s first laser in 1960, organic semiconductor materials have accounted for a large proportion of the developments in this field [4]. Compared with inorganic semiconductor materials, organic semiconductor materials exhibit advantageous features including high photoluminescence efficiency, wide emission spectra, and simple fabrication, making them attractive as laser gain materials [5]. Organic semiconductor materials can be divided into organic semiconductor small molecular dyes and organic semiconductor polymers [6,7,8]. Lasers using semiconductor conjugated polymers can be prepared from a solution through different ways. In addition, the luminescence efficiency of conjugated polymers is not severely affected by the quenching effect of the concentration, and the luminescence efficiency of the film state is almost the same as that of the solution state. However, the emission efficiency of organic semiconductor small molecular dyes drops significantly when they occur in high concentrations in a solid laser.

Organic semiconductor polymers with large conjugated systems (conjugated polymers) are a type of polymer that has a large delocalized π bond in the main chain. Conjugated polymers’ main chains are composed of C–C single bonds, double bonds, triple bonds, various aromatic rings, and other unsaturated bonds. The valence of carbon in the repeating unit on the main chain of conjugated polymers is unsaturated; only three of the outermost four free electrons of the C atom are combined with other atomic bonds and are localized in the δ orbital. The unpaired electron is called the π electron. The π electron is delocalized in the Pz orbit, which is perpendicular to the plane of the molecule. A large π bond is formed when the Pz orbital of one carbon atom and the Pz orbital of another carbon atom overlap each other. The semiconductor properties of conjugated polymers are due to the π electrons localized along the polymers. In conjugated polymers, the σ bond is the domain bond that forms the skeleton of the molecular structure; the off-domain π electrons move inside the molecular framework, forming the off-domain π bond that increases the electron activity range, causing the energy level of the polymer to decrease. It has been found that conjugated polymers can easily produce electron energy level transitions under the excitation of light. Moreover, they provide good charge transport, which can achieve stimulated emission amplification in a solid state, and the structures of the conjugated polymers are various and can satisfy various uses, indicating the possibility of laser action in conjugated polymer-based materials [9]. Also, they are easily integrated into various kinds of substrates—for example, through spin-coating to obtain a film with good uniformity. Based on the above advantages, conjugated polymer materials have attracted more and more attention as laser gain materials (Figure 1) [10].

After Burrough of the Cavendish Laboratory at the University of Cambridge in the UK published a paper entitled “Polyphenylene Ethylene (PPV)-Based Single-Layer Electroluminescent Devices” in *Nature* in 1990, conjugated polymers with high luminescence efficiency have constantly emerged [11]. In the development of polymer lasers, many scientists have made remarkable contributions. For example, in 1992, the research team of Moese at the University of California reported the photoinduced radiation phenomenon in a poly(2-methoxy-5-(2′-ethylhexyloxy)-1,4-phenylene vinylene) (MEH-PPV) solution [12]. The quantum efficiency of this material system is similar to that of the traditional dye Rhodamine 6G. Also, the research team of Brouwer at the University of Groningen demonstrated a laser emitted from a poly[(2,5,-tetraoctyl)-*p*-terphenyl-4,4-ylene vinylene-1,4-phenylenevinylene] (TOP-PPV) n-hexane solution in 1995 [13]. The emission wavelength of this material could be adjusted between 414 nm and 456 nm, and its laser efficiency was nearly double that of coumarin 12 and coumarin 47. In 1996, Heeger’s group in the United States mixed poly(2-methoxy-5-(2′-ethylhexyloxy)-1,4-phenylene vinylene) (MEH-PPV) into polystyrene (PS) substrates and added an appropriate amount of TiO_2_ nanocrystals to make films with a micron-scale thickness. Gain narrowing was observed under the pump of a 532 nm pulsed laser (with a threshold of about 1 mJ) [14]. Friend’s team at the Cavendish Laboratory at the University of Cambridge in the UK gave a detailed report on the lasing behavior of optically pumped polymer microcavity lasers in the same year. Since then, several research groups around the world have independently reported the phenomenon of stimulated emission or laser amplification in polymer films at almost the same time, publishing important academic papers in the journals *Nature* and *Science*, respectively. This paved the way for further research of conjugated polymer lasers [15,16,17,18]. In 2003, Samuel et al. demonstrated a surface-emitting conjugated polymer laser, this compact and convenient laser system has the potential for spectral tuning [19]. In 2007, Redmond et al. reported the first incidence of optically pumped lasing in conjugated PFO nanowires which were fabricated through the template wetting method in a single Fabry–Pérot mode. The laser appeared above the energy threshold of 100 nJ (2.8 mJ·cm^−2^) [20]. In 2013, Herrnsdorf et al. reported the organic solid-state laser using conjugated polymer poly [2,5-bis(2,5-bis(2-ethylhexyloxy)phenyl)-*p*-phenylenevinylene] (BBEHP-PPV), which could be integrated with light-emitting diodes for hybrid laser devices [21]. In 2015, Zhang et al. fabricated a tunable multi-wavelength polymer laser through spin-coating the conjugated polymer onto a polymer film. By blending the bottom film, they achieved three laser wavelengths due to the change of the scalene triangular compound cavity [22]. Furthermore, they cascaded red-, green-, and blue-light-emitting conjugated polymer membranes and constructed a multilayer structure emitted red–green–blue laser emission. This technique can be used for compact, integrated polymer laser sources [23]. The latest works on conjugated polymer lasers come from 2018. In 2018, Xiao et al. demonstrated an optofluidic microlaser using a biocompatible conjugated polymer (poly[2-methoxy-5-(2- ethylhexyloxy)-1,4-(1-cyanovi-nylene-1,4-phenylene)]) (CN-PPV) as gain materials, due to the electrophilic CN-group substituent, the photostability and lasing stability of the whole conjugated polymer were improved greatly compared with a typical dye laser [24].

The following international research groups have been working on conjugated polymer lasers, following is a list of their works in the past three years. (1) M. S. AlSalhi studied how the different parameters change (including concentration, temperature, solvent type, laser power excitation, etc.) affect the spectral and laser properties of a conjugated polymer, demonstrated the existence of dimeric and excimeric states of conjugated polymer [25], and investigated the spectral and temporal profile of conjugated polymer dimer laser by employing a picosecond spectrometer. They also reported a laser from new copolymer (poly[(9,9-dioctyl-2,7-divinylenefluorenylene)-alt-*co*-{2-methoxy-5-(2-ethylhexyloxy)-1,4-phenylene}](PFO-*co*-MEH-PPV) [26], poly[(9,9-dioctylfluorenyl-2,7-diyl)-*co*-(2,5-*p*-xylene)] also known as [(PFO-*co*-pX), ADS145UV]), studied the relaxation oscillation and the excited state dynamics during laser generation [27], demonstrated the influence of steric hindrance of co-polymer segment on the laser, also these new kinds of conjugated polymers exhibited a violet laser [28]. Furthermore, they fabricated a fully reversible temperature-tunable laser based on the temperature sensitivity conjugated polymer poly[3-(2-ethyl-isocyanato-octadecanyl)-thiophene] (TCP) [29]. (2) R. H. Friend does a lot of work on conjugated polymer blend laser systems, accompanied with the Forest Energy Transfer Process which could lower the laser thresholds between the energy-donating and accepting conjugated polymers [30,31]. (3) The group of D. D. C. Bradley works mainly on how to improve the stability and laser properties of conjugated polymers. They synthesized four kinds of polyfluorene (PF) derivatives with varying degrees of steric effect, studied the influence of steric hindrance on the optical gain properties for polymer laser and found that the steric-hindrance unit is an effective way to improve the spectral stability of a conjugated polymer laser and lower the laser threshold [32]. (4) The research group of I. D. W. Samuel has performed work on the miniaturization and lightness of a conjugated polymer laser. In 2018, they made extremely thin (< 500 nm) and ultralow-weight distributed feedback laser using the conjugated polymer poly(9,9-dioctylfluorene-*co*-benzothiadiazole) (F8BT) with good mechanical flexibility. This ultra-thin polymer membrane laser can be transferred onto a different substrate which could be applied as a security label [33]. These research groups and scientists have greatly advanced the development of conjugated polymer lasers. Now conjugated polymer lasers are often used as visible light sources, they could offer a tunable laser emission that covers the whole visible spectrum. Also the conjugated polymer lasers are used in the field of spectroscopy, sensing, displays and some areas of data communications. Conjugated polymers are inexpensive, easy to be processed and shaped, and can make the laser structure compact and realize the miniaturization of devices. However, a series of challenges remain to be addressed for conjugate polymer laser materials. For example, the stability of the materials needs to be improved, various parameters of conjugated polymer lasers need to be regulated in order to adapt to different application requirements [34,35], and the assembly process of laser devices based on conjugated polymers remains to be studied. Furthermore, conjugated polymer lasers under an electric pump have not been realized to date.

## 2. Laser Process of Conjugated Polymers

As laser gain media, conjugate polymers need to meet the following conditions. First, conjugated polymers must have appropriate energy level systems and be able to achieve particle number inversion between ground and excited states. Second, they should have high luminous efficiency and high chromophore density. Studies have shown that if the conjugated polymer side chains are not intertwined, the photoluminescence efficiency can be as high as 60%, even in undoped films. Third, the excited absorption spectra cannot overlap with the stimulated emission spectra to reduce the self-absorption.

### 2.1. Photoluminescence Principle of Conjugated Polymers

The ground state refers to the stable state of a molecule in photophysics and photochemistry—that is, the lowest state of energy. When electrons in a molecule are excited, the molecule is in the excited state, which is an unstable state with relatively high energy. After absorbing the energy from the external excitation, the electrons jump to the excited state from the ground state, and the luminescence of conjugated polymer materials is the radiation that is generated from the electrons’ return to the ground state from the excited state. Electrons exist in pairs in each molecular orbital in the ground state. According to the Pauli exclusion principle, two electrons in the same orbital have opposite spin, so when the total electron spin in a molecule is zero the molecule is in the ground state (S_0_). When an electron in a molecule is stimulated after absorbing energy, often its spin is unchanged, so the molecule is in the excited singlet state (S). If the electron spin is reversed during excitation, the molecule is in the excited triple state (T). When the molecules of the conjugated polymers are excited from the ground state S_0_ to the excited state S under external excitation, and then relaxed to the lowest level of excitation (that is, the singlet state S_1_) due to the level of vibration, the vibration relaxation process is around 10^−l4^ s~10^−12^ s. Finally, the electrons transit back to the ground state S_0_ from S_1_, and thus they radiate photons. The S_1_→S_0_ process is called the fluorescence, and the time required for fluorescence radiation transition of the excited singlet state is commonly 10^−8^ s [1]. It is also possible that when the energy of the excited state is close to the ground state, the electrons can pass through the barrier between them in a tunneling effect. Thereafter, the molecule is in a highly excited vibrational energy level of the ground state—this process is commonly referred to as an “internal conversion”. Internal conversions, which are competitive with fluorescence radiation, are generally particularly fast (within 10^−11^ s). The above discussion concerns the process of fluorescence radiation. If the electron passes from the lowest excited singlet state S_1_ to the lowest excited triplet state T_1_ by intersystem crossing, and then emits a photon when it returns from T_1_ to S_0_, it is called phosphorescence [36,37]. The transition rate of the phosphorescence process is much smaller than that of the fluorescence process due to the fact that the phosphorescence process is a transition involving the change of spin multiplicities and it is limited by the spin factor. The corresponding phosphorescence life is relatively long, generally within the order of milliseconds or even reaching the order of seconds. The specific process is shown in Figure 2.

### 2.2. Stimulated Radiation Mechanism of Conjugated Polymers

According to the Bohr model, atoms are made up of nuclei and extranuclear electrons, and the orbital radius of the electrons around the nucleus can only be a discrete value. Electrons are in different states when they move in different orbits. Also, atoms have different energies and can only be in a series of discontinuous energy states—these quantized energy values are called energy levels. The energy levels of an atom are also discrete; the lowest energy level is called the ground state and the other energy level is called the excited state (Figure 3). If an electron of an atom jumps from a lower energy level to a higher energy level, it is an absorption process. When the electron jumps from a higher energy level to a lower energy level, it releases a photon with the energy of hν—this is referred to as the emission process. The energy of the photon is equal to the energy difference between the two energy levels, which can be expressed as:hν = E_2_ − E_1_
where E_2_ is the energy of a higher energy level and E_1_ is the energy of a lower energy level.

When a particle at a lower energy level is excited (that is, it interacts with other particles associated with the energy exchange, such as inelastic collisions with photons) and absorbs energy, it transits to a higher energy level corresponding to this energy, which is called stimulated absorption. Particles are stimulated into the excited state, which is not steady. If a lower level that can accept the particles exists, even in the absence of external interaction, particles also have a certain probability of jumping to a lower level ground state (E_1_) from the high excited state (E_2_) level spontaneously. In this situation, they would radiate photons with the energy of (E_2_ − E_1_) at the same time, and the photon frequency ν would be equal to [(E_2_ − E_1_)/h]. This process of radiation is called spontaneous radiation. The light emitted by many atoms in spontaneous emission is called incoherent light, and is not consistent in the phase, polarization, or direction of propagation.

In 1917, Einstein pointed it out that theoretically, in addition to spontaneous radiation, particles on a higher E_2_ energy level could jump to a lower energy level in another way. An incident photon with a frequency of ν (ν = (E_2_ − E_1_)/h) would also cause the particles to rapidly transit from the higher energy level E_2_ to the lower energy level E_1_ with a certain probability, and they would radiate a photon with the same frequency, phase, polarization, and propagation direction as the external photon at the same time. This process is called stimulated radiation. At some states, a weaker light can trigger a stronger light—this is called stimulated light amplification, abbreviated as laser [2]. The processes of the stimulated absorption, spontaneous emission, and stimulated emission of atoms are shown in Figure 4.

For laser gain media, there exists a special energy level. In this energy state, the atom has a relatively high stability and can easily accumulate excited atoms or other particles, which provides the basic conditions for the realization of particle number inversion. Such energy levels are called metastable energy levels. The materials which can realize particle number inversion between the upper and lower levels can be divided into three-level structures and four-level structures (Figure 5). According to the movement of atoms, the three-energy level system is divided into the ground state, the excited state, and the metastable state. After the laser working substance is motivated, the particle is triggered from the ground state to the excited state and jumps to the metastable state. There it stops for a short time (about 10^−3^ s) because the life of the excited state is very short (about 10^−7^ s). With the enhancement of excitation and the passage of time, the number of particles in the metastable state will become greater than the number of particles in the ground state—this state is called the particle number inversion. In the setting of an optical resonator, optical oscillations and the amplification of stimulated radiation can be formed and eventually a laser can be output. In a three-level system, the efficiency of obtaining the particle number inversion between the intermediate energy state and the ground state is not very high. Because the intermediate energy state (i.e., the metastable state) is empty at the beginning of pumping, it is necessary to transfer half of the number of ground state particles to the intermediate state to realize particle number reversion. The four-level energy system can be divided into the ground state, excited state, metastable state, and final state. When the particles are excited, they jump from the lower energy level E_0_ to the higher energy level E_3_, and E_2_ is the upper energy level of the metastable state. The particles excited to E_3_ are rapidly transferred to the metastable state through non-radiative transition. The particles can accumulate at E_2_ due to the long metastable life and realize particle number inversion between E_2_ and E_1_ [38]. For a four-level system, because the E_1_ is not the ground level but an excited level, and is basically empty at room temperature, the number of particles in the final state is very small. As long as the number of particles in the metastable state is greater than the number of particles in the final state, particle number inversion can be realized. Therefore, the excitation energy of a four-level system is much smaller than that of a three-level system and can produce a laser far more easily than a three-level system [39,40].

There are two electron energy levels in conjugated polymer materials: the low level (ground state S_0_) and the high level (first excited state S_1_). Each electron energy level is composed of many vibration energy levels, with an energy interval of about 0.2 ev between them. When the absorption energy of conjugated polymers is greater than the width of the forbidden band energy, the electrons are excited from the lowest vibrational level in the ground state to the higher vibrational level in the excited state. They return to the lowest vibrational energy level in the excited state by non-radiative relaxation, then reach to the higher vibrational energy level in the ground state accompanied with emitting light, and finally return back to the lowest vibrational level of the ground state by fast non-radiative relaxation. Since the rate of non-radiative relaxation is very fast (generally three orders of magnitude less than the energy level lifetime), the number of particles accumulated at the lowest vibrational level of the excited state may exceed that at the higher vibrational level of the ground state, so particle number inversion is achieved between these two energy levels. In this way, a typical four-level laser system is formed, which provides the basic condition for the realization of stimulated radiation. When photons of spontaneous radiation induce electrons of the same energy to jump down to the ground state and simultaneously emit two photons of the same mode, this process is called stimulated radiation. Photons of stimulated radiation that can induce excited electrons to jump down, emitting more photons and resulting in the stimulated radiation light amplification—namely, a laser output [3,41] (Figure 6).

It can be seen from the above analysis of the laser emission from the conjugated polymer that the fluorescence spectrum of the conjugated polymer materials undergo an obvious redshift in the absorption spectrum due to the radiation relaxation of electrons in the excited state, which is called Stokes shift. For conjugate polymer laser gain materials, a large Stokes shift is a positive factor, because it can reduce the self-absorption of the materials, thus reducing the loss and improving the luminous efficiency. For many other materials, electrons from the excited state are likely to jump to the higher level or triplet state. The triplet state has a long life which reduces the number of molecules in the excited state. This is disadvantageous to the establishment of particle number inversion; therefore, the problem of the triplet state cannot be neglected. However, for most conjugated polymers, triplet state absorption is not serious. Based on the unique physical and chemical properties listed below, conjugated polymer materials have many advantages as laser gain materials.
(1)High laser emission efficiency. The absorption coefficient is high (α_max_ > 10^5^ cm^−1^) and the stimulated emission cross-section is large (about 10^−15^ cm^2^) [42]. This is due to the large Stokes shift between the absorption spectra and the fluorescence emission spectra of the conjugated polymers. The self-absorption coefficient of the molecule is small, but the absorption coefficient of the excited light is high, and the stimulated radiation has a large advantage relative to spontaneous radiation. For example, a 150 nm film can absorb more than 90% of pump photons, which helps to lower the operating threshold. Also as mentioned above, the laser efficiency of conjugated polymers is not severely affected by the concentration quenching effect, so the conjugated polymer materials still have high luminescence efficiency in the solid state.(2)Easy to achieve particle number inversion. The molecule has a conjugated π–π* bond structure. Moreover, direct transitions between the bonds have a high density of intersections, so they undergo particle number inversion at very low pump intensity (<1000 W/cm^2^). The binding between the molecules relies on van der Waals forces that make the overlap of the electron clouds very small. Furthermore, the carriers are highly localized.(3)Sources of materials are abundant. Different wavelengths of emitted light can be obtained by adjusting the chain length of the conjugated polymer and modifying the groups of the main chain. The spectra range covers the whole visible light region. The wide spectrum also means that the wavelength of conjugated polymer lasers can be modulated in a wide range, allowing different application requirements.

One of the negative aspects that affects the stimulated emission stability of conjugated polymers is photooxidation [43]. The results of the Cambridge research group suggested that species or defects produced by photooxidation can not only effectively quench the singlet excitons, but can also directly or indirectly produce very strong photon avalanches. It is generally believed that photooxidation has a significant negative effect on conjugated polymers. In order to obtain stable laser gain, photooxidation of the material must be absolutely avoided. There exist some ways to improve the stability of conjugated polymer laser materials, for example, Tang et al. investigated the photostability of conjugated polymer microlasers. Compared with traditional dye laser materials, the conjugated polymer substituted with electrophilic cyano is much more photostable [24].

### 2.3. Effect of Conjugated Polymer Structures on the Laser Threshold

In terms of laser gain materials, conjugated polymer laser materials have become a research hotspot. How to change the structure of conjugated polymers to improve the luminescence efficiency and reduce the threshold of the laser has been the focus of much research.
(a)Effect of the conjugate chain length. The bandgap in conjugated polymers is determined by the degree of π-delocalization along the backbone—the so-called effective conjugation length. The different numbers of unsaturated double bonds and aromatic rings in the molecule lead to different conjugate degrees and molecular plane degrees, which result in different luminescence efficiencies and different luminescence wavelengths. Generally, the luminescence intensity of conjugated polymers which contain aromatic rings or aromatic complex rings is large. This is because a larger conjugate system will cause delocalized electrons to be excited more easily, resulting in a higher quantum efficiency.(b)Effect of substituents. If a molecule contains some groups that increase the luminescence efficiency, these groups called chromophores. Chromophores are generally electron donors (e.g., -NH_2_, -NHR, -NR_2_, -OH, -OR, -CN, etc.). Polymers containing chromophores have unbonded lone pair electrons (known as n electrons), and n electrons’ clouds can be almost parallel to the π orbital on the aromatic ring, so they actually share the electron structure of the π conjugated electrons. Furthermore, they expand the conjugated system, so the luminescence efficiency of these conjugated polymers increases.(c)Effect of space. A large number of studies have found that conjugated polymers with a relatively rigid planar structure have a more stable homogeneous conjugated system and a higher luminescence efficiency. This is mainly due to the reduction of internal conversion probability caused by vibration dissipation [1,15].

## 3. Types of Conjugated Polymer Laser Materials

Recently, many kinds of optical amplification and laser phenomena of conjugated polymers have been reported. Among them, the first observation of stimulated emission and the most studied example is the occurrence of poly(phenylenevinylene) (PPV) derivatives, such as poly[2-methoxy-5-(2′-ethylhexyloxy)-1,4-phenylenevinylene] (MEH-PPV) and poly[2,5-bis(2′-ethylhexyloxy)-1,4-phenylenevinylene] (BuEH-PPV) [44]. Other polymers such as poly(*p*-phenylene) (PPP) derivatives [45,46], polyfluorene (PF) derivatives [47,48], and so on, have also been demonstrated as laser gain media with high efficiencies, in which the spectral range spans the whole visible spectrum from blue to red light.

### 3.1. PPV and PPV Derivatives

At present, the most extensive and in-depth study of conjugated polymer laser materials has been focused on PPV and its derivatives, which have a relatively high brightness, easy band gap control, and a low threshold for gain narrowing [14]. PPV is a typical linear conjugated polymer material and its derivatives have a higher molecular weight, enabling the formation of a high-quality film. Numerous reports have described the laser emission from PPV and its derivatives with high efficiencies at low pump thresholds. Completely conjugated polymers without substituents are insoluble; however, by introducing flexible side chains such as alkanes (oxygen) to the main chain of PPV, the solubility of the polymer can be increased, the stability of the material can be improved, the length of the conjugate chains can be effectively controlled, the luminescence wavelength of the polymer can be changed, and the quantum efficiency of luminescence can be improved. In addition, the absorption spectrum of PPV is strong and wide, so when excited, a 150 nm thick PPV film can absorb more than 90% of the excited photons. The most studied PPV derivatives are MEH-PPV and BuEH-PPV [49], and others include BEH-PPV, BCHA-PPV [50], TOP-PPV [51], DP6-PPV, and so on. The chemical structures of some PPV derivatives are shown in Figure 7. Holzer et al. studied the properties of various PPV copolymers and PPV derivatives whose hydrogens are substituted on the double bonds. They found that the wavelength could be selected or modulated by adjusting the properties of the substituent groups and the content of the copolymers [51].

### 3.2. PPP and PPP Derivatives

The derivatives of *p*-phenylene (PPP) are all conjugated polymers that emit blue light (Figure 8). Their stimulated emissions in the spectral region from 480 to 520 nm do not compete with photoinduced absorption spectra, which make them very appealing as active blue light laser media [52,53]. This kind of material is relatively stable with an energy gap of nearly 3 eV, which meets the requirement of a blue light-emitting material. The first stimulated emission of undoped conjugated polymers was found in methyl-substituted ladder-type poly(*p*-phenylene) (MeLPPP) [54]. Since then, with the use of new cavity structures (two-dimensional optical crystals, circular gratings, etc.), laser emission in PPP films has been reported frequently.

### 3.3. PF and PF Derivatives

Poly(9,9-dioctylfluorene) (PF) derivatives have rigid planar structures and are easy to introduce into flexible alkyls, with excellent solubility [55]. The structures of common PF derivatives are shown in Figure 9. Generally, they have very high fluorescence quantum efficiencies. In particular, their blue laser emission efficiencies and solid-state fluorescence quantum efficiencies reach up to 60%~80% [56,57,58]. The first report of PF as laser material appeared in 1997. Müller et al. believed that the spectrum line narrowing phenomenon in PFO was the result of the combined action of amplified self-emission and superfluorescence [59]. Lan et al. calculated the excitation parameters of the PFO copolymer, showing that its stimulated emission cross-sectional area at 448 nm was about 1.6 × 10^−16^ cm^2^, and its absorption cross-sectional area at 370 nm was about 2.8 × 10^−16^ cm^2^. When pumped with 1.4 kW/cm^2^, the net gain coefficient was 26 ± 1.7 cm^−1^ and the loss coefficient was 13 ± 1.1 cm^−1^ at the peak of ASE [60].

### 3.4. PT and PT Derivatives

Polythiophene (PT) and its derivatives are also important heterocyclic conjugated polymer luminescent materials (Figure 10). As there are many modifiable positions on polythiophene which can lead to different spatial configurations, the energy gap can be adjusted by changing the side chain of the polymer to obtain different color luminescent materials. In 1998, Granlund et al. reported a photopumped lasing phenomenon with PT as the emitter. They spin-coated the PT solution on top of the dielectric mirrors, pressing them at elevated temperatures to prepare optically flat films. The PT film was excited at a low pump power and the photoluminescence quantum yield was 24% [61].

### 3.5. Copolymer

In addition to polymers formed by a single monomer unit, copolymers formed by different monomer units are also widely used as laser gain materials. Copolymers can combine the advantages of different polymers and form high-quality bulk laser media. Energy transfer, which has been proved to be an efficient way to achieve low-threshold lasers, can occur between different monomers in copolymers, because the absorption and emission spectra are separated. Theander et al. synthesized a series of copolymers of PF and PPV, as shown in Figure 11. The interchain excitons in pure PF were suppressed in the copolymer, so the whole copolymer could be used as highly luminescent laser material. They studied the energy transfer phenomena and found that there energy transfer from PF to PPV occurred in both solutions and films, thus avoiding the overlapping of the absorption and emission spectra. After the microcavity structure model was established, the laser threshold was found to be 2 uJ/cm^2^ [62].

## 4. Conclusions and Future Developments

To sum up, conjugated polymer materials, due to their microcosmic four-level structure and appreciable Stokes shift characteristics, as well as their macroscopic preparation cost advantages and ease of processing, are not only in line with the theoretical conditions for the realization of laser emission, but are expected to become ideal laser materials in solutions and solid blends with broad application prospects, opening the way towards new laser actions based on organic polymer materials.

Polymer lasers under the condition of a light pump have been fabricated, but it is still a very challenging problem to obtain polymer lasers under an electric pump. The electro-pump organic solid laser is widely recognized as the most important way to fabricate a new generation of ultra-portable, cheap, tunable, and flexible laser devices. However, such an electro-pump organic laser has not yet been realized. Compared to the optical pump, the electric pump has a higher additional loss in laser emission. Some of the sources of this loss include the quenching of the electrode on the exciton, the additional absorption of the electrode on the photon, the carrier (dipole), and the quenching of the triplet state. The main challenge includes the following two aspects:(1)To reach the pump emission threshold, the working current density with electromechanical luminescence is usually at about 10^5^ A/cm^2^, while the current density in the conjugated polymers is about at 10^2^ A/cm^2^, with a difference of more than three orders of magnitude [63]. To achieve such a high current density, the carrier mobility and thermal stability of conjugated polymer materials must be greatly improved, otherwise the joule heat generated by high voltage will cause damage to the organic materials and result in the invalidation of the device.(2)In an electric pump structure, the conjugated polymer materials must be embedded between a pair of positive and negative poles. On the one hand, the electrode will produce a great deal of light loss [64]. An even more important aspect is that organic polymer materials must use a grating laser resonant structure (a structure of several hundred nanometers in size) which will largely crack the electrodes deposited on them, reduce the ohm characteristics of electrical contact, and affect carrier injection. Therefore, the effect of the resonant cavity structure on the electrode mass must be considered when designing a conjugated polymer laser structure.

It is not easy to solve these problems, but the research progress in this field is brimming with potential. Once conjugated polymer electro-pump lasers can be realized, it will greatly promote the application of laser devices and the development towards miniaturization, so as to meet the demand for new materials in the information age.

## Figures and Tables

**Figure 1 polymers-11-00443-f001:**
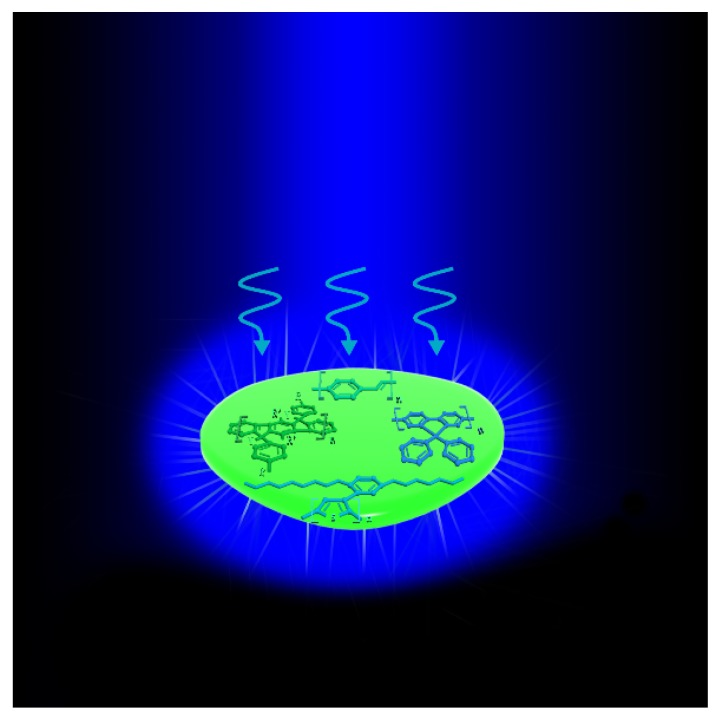
Schematic diagram of conjugated polymer lasers.

**Figure 2 polymers-11-00443-f002:**
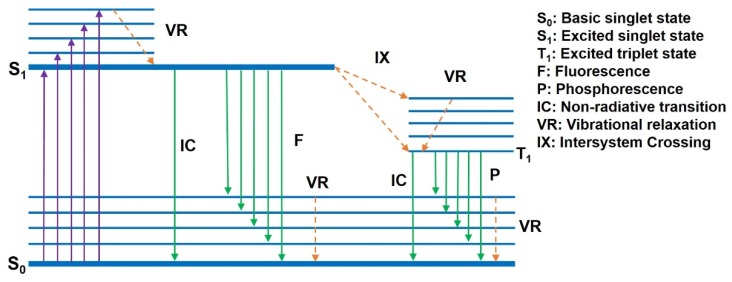
Schematic diagram of radiative transition for typical conjugated polymer materials.

**Figure 3 polymers-11-00443-f003:**
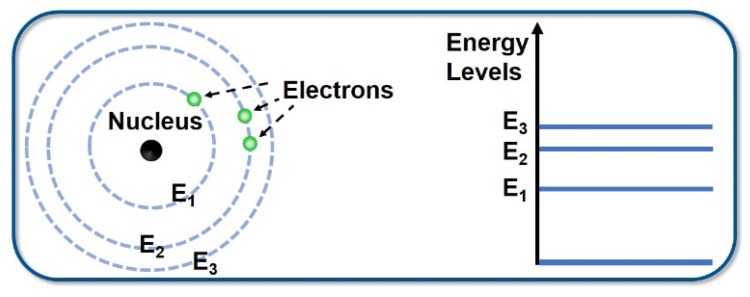
Transition level diagram of electrons.

**Figure 4 polymers-11-00443-f004:**
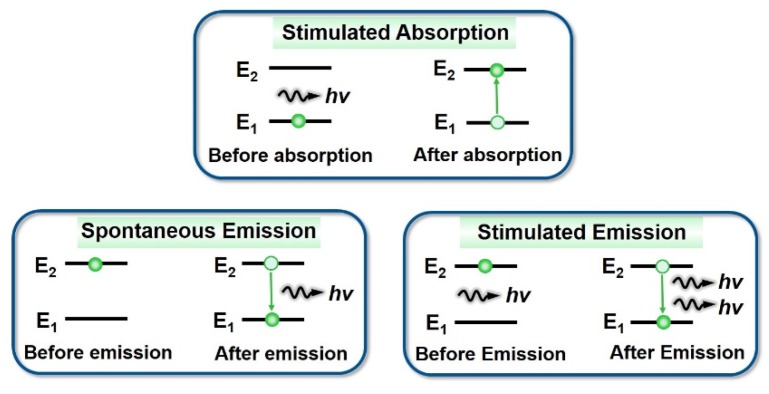
Processes of stimulated absorption, spontaneous emission, and stimulated emission.

**Figure 5 polymers-11-00443-f005:**
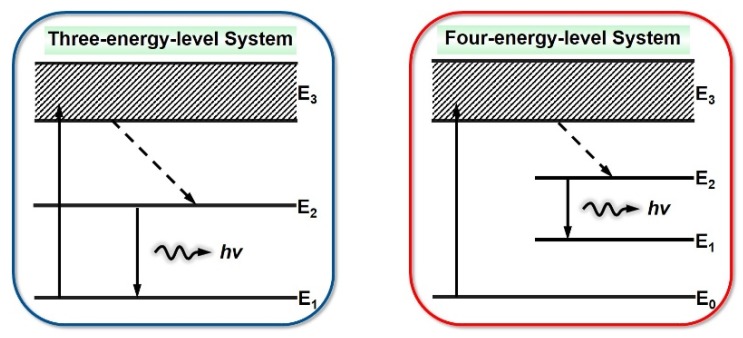
Energy level diagram of three-level laser materials and four-level laser materials.

**Figure 6 polymers-11-00443-f006:**
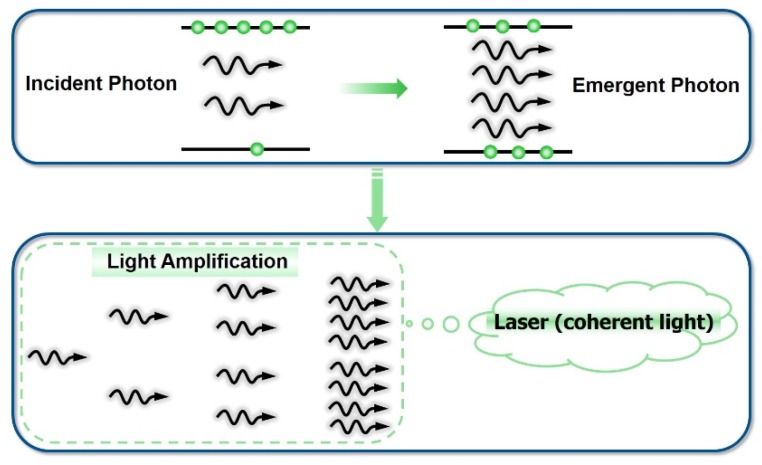
Process of light amplification.

**Figure 7 polymers-11-00443-f007:**
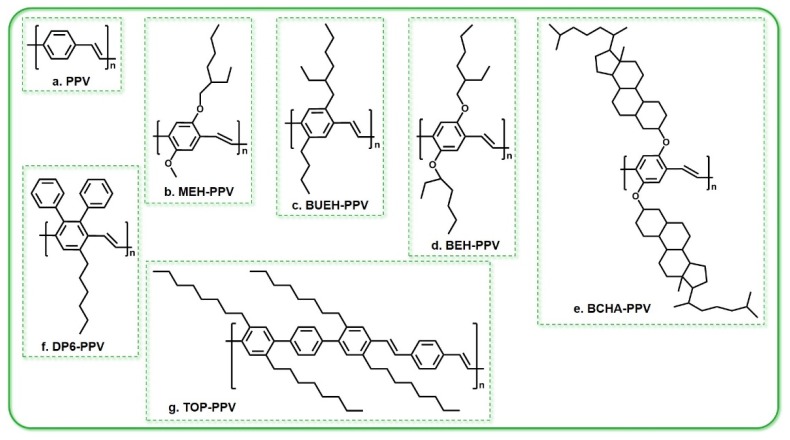
The chemical structures of some PPV derivatives. (**a**) PPV: poly(phenylenevinylene); (**b**) MEH-PPV: poly[2-methoxy-5-(2′-ethylhexyloxy)-1,4-phenylenevinylene]; (**c**) BuEH-PPV: poly[2-butyl-5-(2′-ethyl-hexyl)-1,4-phenylenevinylene]; (**d**) BEH–PPV: poly[2,5-bis(2′-ethylhexyloxy)-1,4-phenylenevinylene]; (**e**) BCHA–PPV: poly[2,5-bis(cholestanoxy)-1,4-phenylenevinylene]; (**f**) DP6-PPV: poly(2,3-diphenyl-1,4-phenylenevinylene); (**g**) TOP-PPV: poly[(2,5,-tetraoctyl)-*p*-terphenyl-4,4-ylene vinylene-1,4-phenylenevinylene].

**Figure 8 polymers-11-00443-f008:**
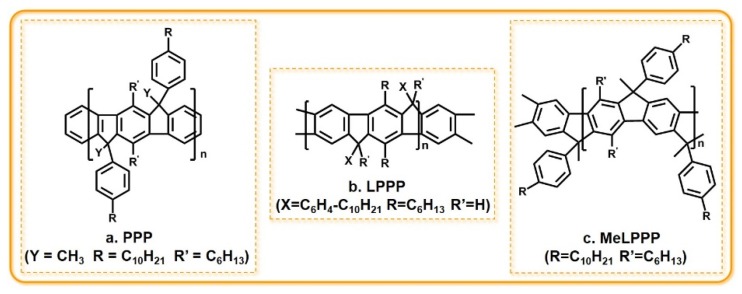
The chemical structure of some PPP derivatives. (**a**) PPP: poly(para-phenylene); (**b**) LPPP: ladder-type poly(*p*-phyenylene); (**c**) MeLPPP: methyl-substituted ladder-type poly(*p*-phenylene).

**Figure 9 polymers-11-00443-f009:**
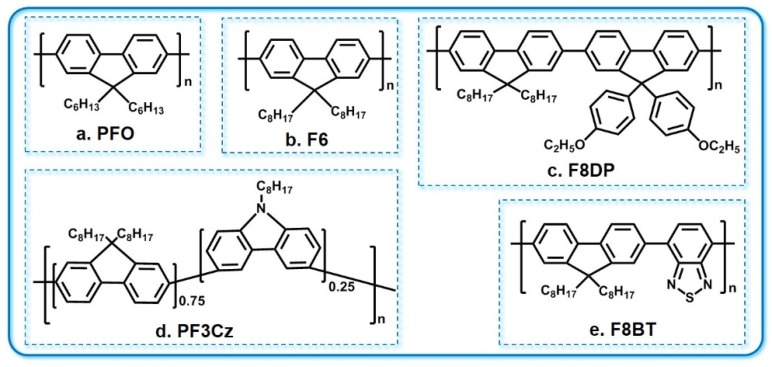
The chemical structure of some PF derivatives. (**a**) PFO: poly(9,9-dioctylfluorene); (**b**) F6: poly(9,9-dihexylfluorene); (**c**) F8DP: poly[(9,9-dioctylfluorene)-*co*-(9,9-di(4-methoxy) phenylfluorene]; (**d**) PF3Cz: poly[(9,9-dioctylfluorene)-*co*-carbazole]; (**e**) F8BT: poly[(9,9-dioctylfluorene)-*co*-benzothiadiazole].

**Figure 10 polymers-11-00443-f010:**
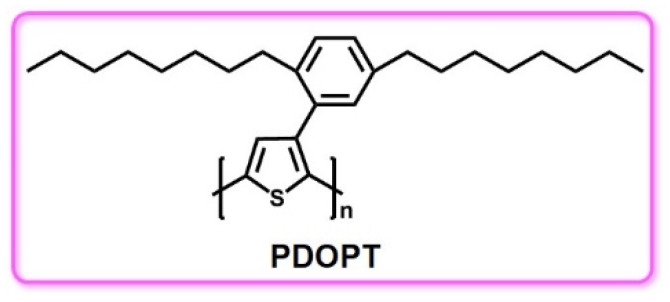
The chemical structure of PDOPT: poly(3-(2,5-dioctylphenyl) thiophene).

**Figure 11 polymers-11-00443-f011:**
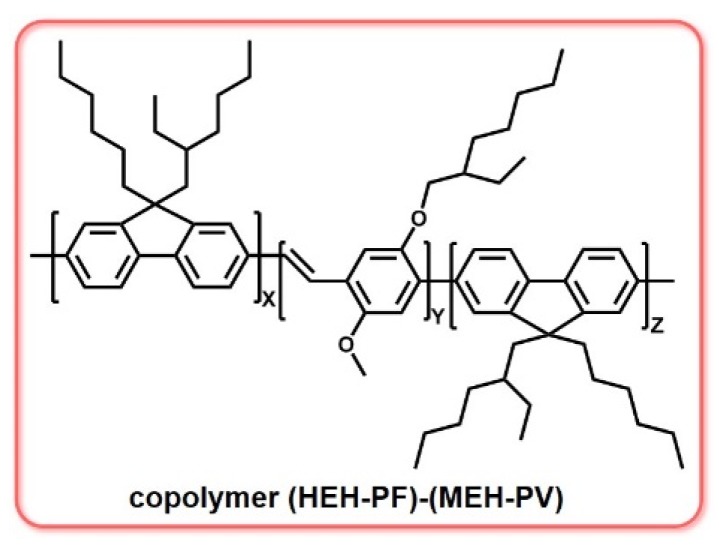
The chemical structure of copolymers of PF and PPV: poly(9,9-methylhexyl-hexyl) fluorene (HEH-PF) and 2-methoxy-5-(2′-ethyl-hexyloxy)-*p*-phenylenevinylene (MEH-PPV).

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
