# Peer review of "Advances in Conjugated Polymer Lasers"

_polymers, 2019, doi:10.3390/polym11030443_

Round 1

Reviewer 1 Report

This is a review on "advances in conjugated polymer lasers" however, at this point it cannot be considered a review since references are not up to date. In fact, there is one reference from 2018, another one from 2015 and then the rest are from 2008 or older. Thus, in order this work to be considered a review first of all references should be updated and state of the art explained. Write a review needs to invest time in order to contain all the relevant work done on the topic. Otherwise if there are no papers published on the topic that would mean this is not a hot topic, whch is not the case. If authors want to publish a review they would need to do a serious bibliographic work and give information on important points like stability and efficiency of these lasers, besides some applications. Additionally, some concepts are not well explained. For instance atoms do not jump, but the electrons jump.

Author Response

Please see the file attached.

Reviewer 2 Report

In this work the authors review the advances in conjugated polymer lasers. The paper introduces the general concept of lasers and summarizes the recent progress of laser processes employing conjugated polymers, with a focus on the photoluminescence principle and excitation radiation mechanism of conjugated polymers. The most common used conjugated polymers and the main challenges remaining to be addressed for conjugate polymer laser materials are discussed.

This is potentially an interesting paper. However, there are several aspects that should be solved in order to recommend it for publication.

The points that should be amended are the following:

1.      In the Introduction, a figure illustrating the conjugation of polymers will help the understanding of the third paragraph.

2.      The terms atom and electron are exchanged throughout the section 2.

3.      Additional references should be included in order to illustrate previous work. For example in point (c) of section 2.3, among others.

Author Response

Please see the file attached.

Round 2

Reviewer 1 Report

Some of the questions arised during first revision have been addressed by the authors however I still insist on my first and most important question:

The point is: is this topic of interest and is it worth to write a review? I guess the answer is "Yes" meaning that there is much work being done on the topic and several groups working on it at the present time, many questions answered, some others still open, etc. However, according to the literatura cited in this work it this is not the case, there are no recent refeences (there are only 5 references corresponding to the last 5 years!!!!). Then… does this topic deserves a review??!! 

Please, check for uptodate references and present a real and complete state of the art. I understand that it is time consuming, but a review needs this work to be done. See for instance:

-Steppert et al, ACS Photonics, (2019),  DOI: 10.1021/acsphotonics.8b01641

- AlSalhi et al, Polymers 10, 470 (2018)

- Mujamammi et al, Polymers 9, 648 (2017)

- Lampert et al, Optics and Laser Technology 94, 77 (2017)

- Alfahd et al, Material 10, 265 (2017)

- Mujamammi et al, Polymers 8, 364 (2016)

- Prasad et al, Polymer 55, 727 (2014) 

Just to cite some

Author Response

Please see the file attached.
